# OpenReview forum: "RCD: Retrieval-augmented Contextual Decoding for Truthful Generation"
_ICLR.cc/2026/Conference — ICLR 2026 Conference Withdrawn Submission_

### Official Review · Reviewer_tmrk · 2025-10-28

**Soundness:** 3
**Presentation:** 3
**Contribution:** 2
**Rating:** 4
**Confidence:** 3

**Summary:**

This paper proposes a new decoding method to improve the truthfulness and informativeness of the generated text. Specifically, they first use a few verified correct samples to construct a grounding space, which consists of the embeddings and logit distributions of these samples’ chunks. Then, taking a question as input, they search the most similar embeddings in the grounding space and modify the original logit distribution using the logit distributions of these embeddings. Experiments on TruthfulQA etc datasets validate the effectiveness of the proposed method.

**Strengths:**

1. Propose a simple but effective method.

**Weaknesses:**

1. The proposed LLM decoding method is only tested on Qwen models.
2. The proposed method requires a few **verified correct answers** to construct the grounding space.
3. There is a typo in Line 154: grouding -> grounding
4. The format of Table captions is incorrect, which should be above the tables.

**Questions:**

1. CAD is one of the baselines in this paper, which requires context for each question. And I wonder in your experimental setup, how do you select the context for each question in CAD?
2. The proposed method requires a few samples (questions with verified correct answers) to form the grounding space. The paper clarifies that this method is generalizable in the sense that even if the target data distribution is different from the ground space distribution, the performance is good. This is validated in Table 2: the ground space is formed using TruthfulQA and the testing data is biographies. Since samples in TruthfulQA are of high quality, I wonder what if the ground space is formed using biographies and the testing data is TruthfulQA?
3. Is there any intuition that in Table 3, why increasing the chunk size won’t lead to significantly improved performance? I have this question because I thought increasing chunks provides more truthful embeddings, which is supposed to induce better performance.

---

### Official Review · Reviewer_RL2T · 2025-10-31

**Soundness:** 2
**Presentation:** 2
**Contribution:** 2
**Rating:** 2
**Confidence:** 3

**Summary:**

This paper proposes a retrieval-augmented, context-aware decoding approach aimed at improving the truthfulness of LLM-generated answers. The authors claim that only 10 annotated examples are sufficient to construct a reference grounding space.

**Strengths:**

The method does not require retraining the target model and claims that it will not have prompt sensitivity, multi-pass generation, or internal model states issues.

**Weaknesses:**

1.	The approach assumes that similar contexts in the embedding space yield similar next-token distributions. Unlike kNN-ICL, where the model can internally filter irrelevant contexts in the prompt, this method heavily depends on retrieved results. Factors such as database representativeness, embedding model alignment, and noise in similarity scores could significantly impact generation quality. The paper should clarify how these risks are mitigated.
2.	While the paper critiques multi-pass generation costs in related work, the proposed method introduces its own overhead through online embedding computation and kNN search. Including computational cost as an evaluation metric would strengthen the analysis.
3.	Benchmarks are limited to QA and Biographies tasks. Adding more diverse tasks—such as math reasoning, coding, summarization, etc.—would make the evaluation more convincing.
4.	Experiments are restricted to Qwen2.5-Instruct models. Evaluating across a broader range of models would better demonstrate generalization.
5.	It improves different aspects in QA and Biographies tasks, which seems inconsistent with the paper’s goal.

**Questions:**

1.	Why was T*I chosen? If T is incorrect, what is the rationale for comparing I?
2.	Why do the results in Table 6 not align with those in Table 7?

---

### Official Review · Reviewer_spjw · 2025-11-01

**Soundness:** 3
**Presentation:** 3
**Contribution:** 2
**Rating:** 2
**Confidence:** 3

**Summary:**

This paper presents Retrieval-Augmented Contextual Decoding (RCD), a method that steers an off-the-shelf LLM towards more truthful answers during decoding, without any fine-tuning or external reranker. During generation, the current context is embedded, and the k-most similar stored contexts are retrieved. Their logits are then softmax-weighted and combined with the model’s own logits using a scaling coefficient. Experimental results demonstrate substantial improvements over several sota baselines.

**Strengths:**

1. RCD operates entirely at decoding time and is training-free, requiring only around 10 verified samples to function effectively.

2. Unlike Retrieval-Augmented Generation (RAG), which concatenates retrieved information directly into the context window, RCD operates in the representation space, injecting contextual information through probability reshaping.

3. The paper demonstrates empirical gains on multiple benchmarks, including TruthfulQA, Biographies, and WikiQA, as well as cross-task generalization from TruthfulQA to Bio.

**Weaknesses:**

1. Although RCD is presented as a retrieval-augmented approach, the paper only includes ICL as a baseline. A broader comparison with other retrieval-based or augmentation methods would provide a more comprehensive evaluation.

2. Numerous recent studies have explored this topic, yet these works are not discussed or compared. Notably, the most recent baseline included dates back to 2024, which limits the timeliness of the evaluation.

3. All reported results are based on Qwen-2.5-Instruct models (3B / 7B / 14B), with no experiments on LLaMA or other LLMs. This makes it difficult to assess whether the observed improvements generalize across model families. Moreover, although RCD is training-free, it relies on a pre-constructed knowledge base, which is model-specific and therefore not transferable to other models.

4. The optimal softmax threshold is reported as 0.01, discarding only tokens with less than 1% confidence. However, as shown in Table 5, model performance varies significantly with different threshold values. Interestingly, a threshold of 10%, which should retain only the most confident tokens, does not yield the best performance. This suggests that RCD is sensitive to hyperparameter choices, which may affect its stability and reproducibility.

**Questions:**

See Weakness

---

### Official Review · Reviewer_mXFN · 2025-11-02

**Soundness:** 2
**Presentation:** 3
**Contribution:** 2
**Rating:** 4
**Confidence:** 3

**Summary:**

This paper proposes a decoding-time intervention method called Retrieval-augmented Contextual Decoding (RCD) for improving the truthfulness of large language model (LLM) generation.
The core idea is to construct a small grounding space consisting of pairs of context embeddings and next-token logits derived from a few annotated truthful examples (as few as ten). During inference, the current decoding context is used as a query to retrieve the top-N semantically similar contexts. Their corresponding logits are aggregated and combined with the model’s own logits to guide generation toward more truthful outputs.
The authors evaluate the method on TruthfulQA, Biographies, and WikiQA, showing moderate improvements compared with several strong baselines such as DoLa, Context-Aware Decoding, Instructive Decoding, and In-Context Learning (ICL).

**Strengths:**

The method does not require model retraining or architectural modification. It can be easily applied to different autoregressive models, which makes it scalable.

The paper demonstrates that even with only ten annotated examples, the proposed approach yields measurable gains, showing strong sample efficiency.

The authors evaluate across three factual generation benchmarks and compare with five competitive baselines. The evaluation also includes cross-task generalization and ablation studies.

**Weaknesses:**

The idea of retrieving similar contexts and aggregating logits is conceptually close to existing paradigms such as kNN-LM, DoLa, and Context-Aware Decoding, and Integrative Decoding. The paper lacks a clear theoretical or algorithmic distinction from these prior works.

Improvements on the benchmarks are modest, often within one to three percentage points, and informativeness even decreases on TruthfulQA. For larger models such as Qwen2.5-14B, the gains are marginal.

The paper does not examine how noisy or irrelevant retrieval affects factuality, nor does it show failure cases or retrieval quality analysis.

The paper does not compare with RAG-style retrieval-augmented generation or other retrieval-based factual decoders.

**Questions:**

N/A

---

### Note · Authors · 2025-11-13

**Comment:**

Thanks to the reviewers for their feedback. I have read and agree with the venue’s withdrawal policy on behalf of myself and my co-authors.

**Withdrawal Confirmation:**

I have read and agree with the venue's withdrawal policy on behalf of myself and my co-authors.